# SF-YOLOv5: A Lightweight Small Object Detection Algorithm Based on Improved Feature Fusion Mode

**DOI:** 10.3390/s22155817

**Published:** 2022-08-04

**Authors:** Haiying Liu, Fengqian Sun, Jason Gu, Lixia Deng

**Affiliations:** 1School of Information and Automation Engineering, Qilu University of Technology (Shandong Academy of Sciences), Jinan 250353, China; 2School of Electrical and Computer Engineering, Dalhousie University, Halifax, NS B3H 4R2, Canada

**Keywords:** object detection, visual tracking, small object, segmentation and categorization, YOLO

## Abstract

In the research of computer vision, a very challenging problem is the detection of small objects. The existing detection algorithms often focus on detecting full-scale objects, without making proprietary optimization for detecting small-size objects. For small objects dense scenes, not only the accuracy is low, but also there is a certain waste of computing resources. An improved detection algorithm was proposed for small objects based on YOLOv5. By reasonably clipping the feature map output of the large object detection layer, the computing resources required by the model were significantly reduced and the model becomes more lightweight. An improved feature fusion method (PB-FPN) for small object detection based on PANet and BiFPN was proposed, which effectively increased the detection ability for small object of the algorithm. By introducing the spatial pyramid pooling (SPP) in the backbone network into the feature fusion network and connecting with the model prediction head, the performance of the algorithm was effectively enhanced. The experiments demonstrated that the improved algorithm has very good results in detection accuracy and real-time ability. Compared with the classical YOLOv5, the mAP@0.5 and mAP@0.5:0.95 of SF-YOLOv5 were increased by 1.6% and 0.8%, respectively, the number of parameters of the network were reduced by 68.2%, computational resources (FLOPs) were reduced by 12.7%, and the inferring time of the mode was reduced by 6.9%.

## 1. Introduction

Object detection aims to identify the location, size, spatial relationship, and classes of specific targets in images or video sequences. It is a cross domain of computer vision, digital image processing, and machine vision. It always had high research and application value in face recognition [1,2], industrial defect detection [3], UAV aviation detection [4,5], traffic and vehicle detection [6], pedestrian detection and counting [7], and other fields.

In recent years, with the update and iteration of high-performance GPUs, the continuous emergence and improvement of various large-scale datasets (such as DOTA [8], ImageNet [9], COCO [10]) and the strong rise of neural networks based on convolutional, object detection algorithms using deep learning techniques have become the mainstream of contemporary object detection technology.

The current common object detection algorithms can be divided into two-stage algorithm and one-stage detection algorithm according to whether candidate regions are generated or not. The representative algorithms of the former are: RCNN [11], SPP(Space Pyramid Pooling)-Net [12], Fast-RCNN [13], Faster-RCNN [14], Mask-RCNN [15], etc. The latter mainly includes YOLO (YOLOv1 [16], YOLOv2 [17], YOLOv3 [18], YOLOv4 [19], YOLOv5 [20], etc.) and SSD [21] algorithm. Although the two-stage algorithms can reach high accuracy, the prolonged time of detection makes it hard to satisfy the real-time requirement in daily object detection scenarios. Because it has the advantages of high precision and fast speed, the one-stage detection algorithm has become the research focus in the study of object detection.

For early versions of object detection algorithms, for example, YOLOv1 [16], YOLOv2 [17], etc., its network structure is often the stacking of multiple convolution layers and the full-connection layer. Only the feature map of a fixed size is calculated, which greatly limited the multi-scale detection ability of the model. On this occasion, the detection accuracy of the algorithm for small targets is not ideal.

In the case of solving the above problems, an effective method is to adopt the image pyramid strategy [22], which can enhance the multi-scale detection ability of the model; however, the high computational cost and inference speed limit the development of the image pyramid strategy in an object detection algorithm.

Based on the aforementioned methods, FPN [23] up-samples the feature map and then fuses it with the corresponding output in the backbone network in this process, and generates multiple new feature maps for final detection. This modification improved the feature expression ability of the algorithm and further improved the effect of the algorithm based on real-time detection and multi-scale detection ability. Meanwhile, FPN adopted the top-down feature fusion path, which also strengthened the small object detection performance of the algorithm.

Due to the excellent performance of FPN, adding a feature fusion network after the backbone network of the model become the mainstream configuration of object detection algorithms. For example, YOLOv3 [18], YOLOv4 [19], and YOLOv5 [20], all use FPN as a feature fusion network. The subsequent NAS-FPN [24], ASFF [25], PANet [26], BiFPN [27], and so on have achieved better results through various feature fusion paths, but their basic ideas are the same as that of FPN.

Through the combination of well-designed backbones and feature fusion networks, the current object detection algorithms can ensure the detection accuracy and real-time capability, and have the ability to detect various scales objects; however, for the specific design, these algorithms often focus on various complex life scenarios. Their original purpose is to guarantee that the mathematical model has perfect generalization and detection ability for targets with different scales, but ignore the detection optimization in some specific scale scenes. In this context, designing a simple, efficient, lightweight, and easy-to-deploy algorithm for small object detection has important study significance.

YOLOv5 is the latest version of the YOLO series algorithms. Based on YOLOv5, this research has invented an improved YOLOv5 algorithm for small object detection with the name SF(Small-Fast)-YOLOv5, which can not only significantly reduce the number of parameters and calculated amount of the model, but also has better performance compared with the original version in the small object detection direction.

The major innovations and contributions of this research are as follows:In the default network structure of YOLOv5, the network layer that originally used to generate the large object detection feature map has been reasonably clipped. While realizing the lightweight model, the computing resources required by the model are effectively released and the speed of the model is greatly improved.Based on the PANet [26] and BiFPN [27], a new feature fusion method (PB-FPN) for small object detection is proposed, which improved the detection performance of the algorithm for small targets.The (SPP [12]) layer at the end of the backbone network is introduced into the feature fusion network and connected with multiple prediction heads to improve the feature expression ability of the final output feature map and further enhance the ability of the algorithm.

The other parts of this paper are arranged as follows: Section 2 introduces the basic idea and network structure of the classical algorithm YOLOv5; Section 3 illustrates the improvement strategy of SF-YOLOv5 in detail; Section 4 presents the experimental environment, dataset selection, experimental result analysis, ablation experiment, algorithm comparison, and algorithm generalization ability test; Finally, Section 5 provides the conclusion, and introduces follow-up work and improvement direction.

## 2. Relevant Work

### 2.1. Some Classical Algorithms

The detection of small objects belongs to the problem of abnormal scale in object detection. Most of the classical algorithms are full-size detection algorithms, that is, the algorithm itself will give priority to ensuring the detection ability of objects of various sizes, and make various optimizations for small object detection on this basis. Common optimization strategies include the using of image pyramids [22] or FPN [23]. Adding more detection heads [4] can also attain a good effect, although it will greatly increase the calculation cost.

At present, common detection algorithms (such as [11,12,13,14,15,16,17,18,19,20,21], etc.) have been introduced. In addition, the latest YOLOv7 [28] (which is still under update) and various improved algorithms based on ResNet [29,30] also have excellent performance.

YOLOv5 is an algorithm with high reliability and stability, and it is easy to deploy and train. At the same time, it is also one of the one-stage detection algorithms with the highest accuracy at present; therefore, in this paper, we choose YOLOv5 for subsequent improvement.

### 2.2. Basic Idea of YOLOv5

YOLOv5 continues the consistent idea of the YOLO series in algorithm design: namely, the image to be detected was processed through a input layer (input) and sent to the backbone for feature extraction. The backbone obtains feature maps of different sizes, and then fuses these features through the feature fusion network (neck) to finally generate three feature maps P3, P4, and P5 (in the YOLOv5, the dimensions are expressed with the size of 80 × 80, 40 × 40 and 20 × 20) to detect small, medium, and large objects in the picture, respectively. After the three feature maps were sent to the prediction head (head), the confidence calculation and bounding-box regression were executed for each pixel in the feature map using the preset prior anchor, so as to obtain a multi-dimensional array (BBoxes) including object class, class confidence, box coordinates, width, and height information. By setting the corresponding thresholds (confthreshold, objthreshold) to filter the useless information in the array, and performing a non-maximum suppression (NMS [31]) process, the final detection information can be output. The process of converting the input picture into BBoxes is called the inference process, and the subsequent threshold and NMS operations are called post-processing. The post-processing does not involve the network structure. The default inference process of YOLOv5 can be represented by Figure 1.

In the details of bounding-box regression, different from the previous version of YOLO algorithm, the process of YOLOv5 can be explained by (Equation 1).
(1)$$\begin{array}{c} \begin{array}{cc} g_{x} & =2\sigma \left(s_{x}\right)-0.5+r_{x}\;\\ g_{y} & =2\sigma \left(s_{y}\right)-0.5+r_{y}\;\\ g_{h} & =p_{h}{\left(2\sigma \left(s_{h}\right)\right)}^{2}\;\\ g_{w} & =p_{w}{\left(2\sigma \left(s_{w}\right)\right)}^{2}\; \end{array} \end{array}$$

In the above formula, set the coordinate value of the upper left corner of the feature map to (0, 0). $r_{x}$ and $r_{y}$ are the unadjusted coordinates of the predicted center point. $g_{x}$, $g_{y}$, $g_{w}$, $g_{h}$ represents the information of the adjusted prediction box. $p_{w}$ and $p_{h}$ are for the information of the prior anchor. $s_{x}$ and $s_{y}$ represents the offset calculated by the model. The process of adjusting the center coordinate and size of the preset prior anchor to the center coordinate and size of the final prediction box is called bounding-box regression. There are five versions of YOLOv5, namely YOLOv5x, YOLOv5l, YOLOv5m, YOLOv5s, and YOLOv5n. The performance of each version is shown in Table 1.

In this paper, for the comprehensive consideration of computing resources, model parameters, detection accuracy, deployment ability, and algorithm practicability, we chose YOLOv5s as the basic algorithm and made subsequent improvements and contributions around it.

### 2.3. Network Structure of YOLOv5

Generally speaking, the network structure of YOLOv5 refers to the backbone and neck.

#### 2.3.1. Backbone

The backbone of YOLOv5 is shown in Figure 2. The main structure is the stacking of multiple CBS (Conv + BatchNorm + SiLU) modules and C3 modules, and finally one SPPF module is connected. CBS module is used to assist C3 module in feature extraction, while SPPF module enhances the feature expression ability of the backbone.

Therefore, in the backbone of YOLOv5, the most important layer is the C3 module. The basic idea of C3 comes from CSPNet (cross stage partial networks [32]). C3 can actually be regarded as the specific implementation of CSPNet. YOLOv5 uses the idea of CSPNet to build the C3 module, which not only ensures that the backbone has excellent feature extraction ability, but also curbs the problem of gradient information duplication in the backbone.

#### 2.3.2. Neck

In the neck, YOLOv5 uses the methods of FPN [23] and PAN [26], as shown in Figure 3. The basic idea of FPN is to up-sampling the output feature map (C3, C4, and C5) generated by multiple convolution down sampling operations from the feature extraction network to generate multiple new feature maps (P3, P4, and P5) for detecting different scales targets.

The feature fusion path of FPN is top-down. On this basis, PAN reintroduces a new bottom-up feature fusion path, which further enhance the detection accuracy for different scales objects.

## 3. Approach

The performance of YOLOv5 algorithm is very excellent for object detection, but when the scenario is full of small targets, the detected results cannot obtain ideal results, so there is still has large room for improvement. In order to solve these issues, some questions were proposed:YOLOv5’s feature extraction network set three different sizes of feature map output for detection scenes of various scales. The process of obtaining the feature map requires multiple convolution down sampling operations, which takes up a lot of computing resources and parameters; however, in the actual detection for small targets, is the down sampling operation and feature fusion process for high-level feature map necessary?The feature fusion network of YOLOv5 combined a top-down path (FPN) with a bottom-up path (PAN), which enhances the detection performance on various scale objects; however, for dense small object scenes, can we set a new feature fusion path to improve the feature expression potential of the output feature map? Can we fuse features horizontally to further enhance the detection performance of the algorithm?YOLOv5 adds an SPPF module at the end of the backbone, which improves the performance of the feature extraction network through multiple convolution and pooling operations with different sizes. Can we use this idea to further tap the feature expression potential of the feature map in the end of the neck?

Aiming at the above three questions, we proposed a novel small object detection algorithm: SF(Small-Fast)-YOLOv5. In the multiple down sampling process of the backbone, in order to bring about the lightweightness of the algorithm, we canceled the default C5 (fifth convolution down sampling operation) layer, and made corresponding adjustments in the feature fusion part, leaving only the C3 and C4 feature maps for feature fusion. In the part of the neck, based on PANet and BiFPN, we redesigned a new small object detection feature fusion path (PB-FPN), which aims to strengthen the fusion effect of the model on the underlying features and strengthen the small object detection performance. Meanwhile, we introduce the SPPF module that originally existed only in the backbone at the connection between the neck and the head, which further tapped the feature expression potential of the output feature maps.

### 3.1. Feature Map Clipping

YOLOv5 outputed three sizes of feature maps by default, which are used to predict the large, medium, and small objects, respectively. The way to obtain these feature maps is to carry out continuous convolution down sampling operations and feature fusion processes. Table 2 shows the numerical visualization results of YOLOv5 default feature extraction network.

As shown in Table 2, C represents the position of the module, From −1 indicates that the input of this layer comes from the previous layer, n indicates the amount of modules, params indicates the parameter usage of this layer, module indicates the name of the module, arguments represent the input channel value, output channel value, and convolution kernel attribute, respectively. The 7 and 8 layers indicates the C5 sampling layer in the feature extraction network; as can be seen, the sampling layer occupies enormous parameters.

In the proposed mathematical model, for the purpose of being lightweight, the default C5 feature map of the traditional YOLOv5 is deleted. The feature map clipping corresponding to the improved method in this section can be represented by the flowchart in Figure 4.

### 3.2. Improvement for Feature Fusion Path (PB-FPN)

Early object detection algorithms usually used the high-level feature map that generated by multiple down sampling operations for detection, which will make the model unable to effectively predict objects of various scales. The introduction of FPN [23] solves this problem well, and its basic process is shown in Figure 5b.

Taking the P4 node in Figure 5d as an example:(2)$$\begin{array}{c} \begin{array}{c} P_{4}^{td}=Conv\left(\frac{w_{1}\cdot P_{4}^{in}+w_{2}\cdot Resize\left(P_{5}^{in}\right)}{w_{1}+w_{2}+\varepsilon }\right)\; \end{array} \end{array}$$
(3)$$\begin{array}{c} \begin{array}{c} P_{4}^{out}=Conv\left(\frac{{w_{1}}^{\prime }\cdot P_{4}^{in}+w_{2}^{\prime }\cdot P_{4}^{td}+w_{3}^{\prime }\cdot Resize\left(P_{3}^{out\;}\right)}{w_{1}^{\prime }+w_{2}^{\prime }+w_{3}^{\prime }+\varepsilon }\right)\; \end{array} \end{array}$$

In the (Equation 2) and (Equation 3), $P_{4}^{td}$ represents the intermediate feature map between C4 and P4, $P_{4}^{in}$ and $P_{4}^{out}$ correspond to C4 and P4 respectively, Conv and Resize correspond to convolution and sampling operation respectively, *w* and ε represent weight and a preset small value to avoid numerical instability, which is usually set to 0.0001.

Based on the previous analysis, we designed an improved feature fusion path, as shown in Figure 6:

Compared with the classical algorithms PANet [26] and BiFPN [27], this section additionally introduces a new feature fusion path to further integrate the features from the high-level into the bottom. Meanwhile, new branches are set horizontally to participate in the fusion process of the bottom, which further enhances the detection effect of the algorithm for small targets.

### 3.3. The Improved Feature Fusion Network (SPPF)

The latest version of YOLOv5 used the SPPF (SPP-Fast) module. Compared with the original SPP module, its structure comparison is shown in Figure 7. By simplifying the pooling process, SPPF avoided the repeated operation of SPP and significantly improved the running speed of the module.

In this paper, we introduced several SPPF modules at the connection between the feature fusion network and the model prediction head, in order to further tap the feature expression potential of the finally output feature map by the neck and sent to the head, and further enhance the performance of the model. The modified network in this section can be expressed in Figure 8.

## 4. Experiments

### 4.1. Experimental Environment

In this experiment, the workstation is win10, 16 GB, the CPU is used AMD-4800H, the GPU is used Nvidia-Rtx2060. The algorithm is based on PyTorch, using CUDA11.1 to operation acceleration. The training epoch is set to 300, and the mosaic data enhancement and a prior anchor adaptive adjustment strategy are used.

### 4.2. Dataset Introduction

In order to testify the performance of the improved algorithm (SF-YOLOv5) for the small object detection, this study adopted the WIDER FACE [33] dataset to train and verify the algorithm. This dataset contains the annotation information of 393,703 faces. As shown in Figure 9, the dataset is highly variable in scale, posture, angle, light, and occlusion. The dataset is complex and includes a great quantity of dense small targets.

Considering that the SF-YOLOv5 mainly detects small objects, while the original WIDER FACE dataset contains not only small targets, but also large targets, which will affect the experimental results; therefore, we first screened according to the number and size of face images in a single image in the original dataset. In total, 4441 images were used to train the algorithm and 1123 images were used to verify the algorithm. The number proportion of the train and verify was 4:1, same as the original dataset. Figure 10 is an attribute visualization result of the new dataset.

Figure 10a expressed the number of labels in the dataset. Figure 10b shows the central coordinate position of the object. Figure 10c shows the size of the object. It can be seen that this dataset is sufficient to ensure the training and verification of small object detection algorithm.

### 4.3. Evaluation Criterion

At present, the mainstream general indicators for evaluating the performance of object detection algorithms include precision, recall, AP (average precision), mAP (mean AP), parameters(the number of parameters in model), FLOPs (floating point operations per second), inference time, etc.

Considering that mAP can more reasonably reflect the accuracy of the algorithm than precision and recall, in this experiment, we choose mAP@0.5, mAP@0.5:0.95, parameters, FLOPs, and inference time to evaluate the algorithm.

### 4.4. Analysis of Experimental Results

The experimental performance comparison between YOLOv5s and SF-YOLOv5 is illustrated in Table 3. Compared with the traditional algorithm YOLOv5s, the mAP@0.5 and mAP@0.5:0.95 of SF-YOLOv5 has been increased by 1.6 and 0.8, respectively, which proved the improvement in comprehensive detection performance of SF-YOLOv5. At the same time, the parameters (M) value and FLOPs (G) value of SF-YOLOv5 are reduced by 68.2% and 12.7%, respectively, indicating that SF-YOLOv5 can further decrease the number of parameters and lessen computing power required for model operation under the condition of improving the detection accuracy, which enhances the deployment performance of the algorithm on various low-end workstations and small mobile devices. The reduction in inference time (ms) shows that the model can process more images and videos in the same time, which ensures the speed of SF-YOLOv5 in detecting small objects.

### 4.5. Ablation Experiment

Table 3 expressed the ablation experimental data of improved methods 1, 2, and 3 proposed in this paper. YOLOv5-N5 is a new algorithm obtained by cutting the feature layer with the default YOLOv5, which corresponds to the improved method 1 proposed in this paper. It is not difficult to find that although this improvement has increased by 0.2 in mAP@0.5, the value of mAP@0.5:0.95 has decreased by 0.2. The calculation formula of GIOU [34] used by YOLOv5 is shown in (Equation 4).
(4)$$\begin{array}{c} \begin{array}{c} L=1-IoU+\frac{\left|A-P\cup P^{gt}\right|}{\left|A\right|}\; \end{array} \end{array}$$
where *P* and $P^{gt}$ express the area of the prediction box and ground truth box, respectively, and *A* is the minimum area of the area composed of *P* and $P^{gt}$. We hope the value of *L* to be infinitely close to 0. In this case, the lower the value of IOU, the lower the requirement for the position coincidence when verifying the accuracy of the algorithm.

It can be seen from Table 3 and formula (Equation 4), the improved method 1 can achieve significant weight improvements after removing the high-level feature map. Although the detection performance has been slightly affected: the model can identify more objects; the position of the prediction box will deviate to a certain extent.

YOLOv5-PB is a new algorithm obtained by setting the feature fusion path as PB-FPN on the basis of the improved method 1, corresponding to the improved method 2 in this paper. Compared with YOLOv5-N5, this algorithm slightly increases the computational resources and number of parameters, but greatly enhances the detection accuracy, the value of mAP@0.5 and mAP@0.5:0.95 by 0.9 and 0.7, respectively. It proves the feasibility of introducing a more efficient feature fusion path to enhance the ability of algorithm.

The final SF-YOLOV5 was obtained by adding SPPF module at the junction between the neck and head of YOLOV5-PB, corresponding to the improved method 3. Compared with the improved method 2, YOLOV5-PB slightly increases the number of parameters and further enhances the detection accuracy of the algorithm.

Figure 11 shows the visualization of ablation experimental data of improved methods 1, 2, and 3. Figure 12 shows the detection results of SF-YOLOv5.

### 4.6. Comparison with Other Classical Algorithms

We compare SF-YOLOv5 with several classical detection algorithms, and the results are expressed in Table 3. Among them, YOLOv5n is the latest lightweight algorithm of YOLOv5, and YOLOv3 is a relatively mature large-scale one-stage detection algorithm. YOLOv7 [28] is the latest algorithm of YOLO family at present, which has the strongest comprehensive performance in full-scale detection, and YOLOv7-tiny is a lightweight version of YOLOv7, which has similar parameter quantities and calculation quantities with SF-YOLOv5. ResNeXt-CSP is a new detector combined with classical algorithms ResNeXt [29] and CSPNet [32], which have excellent performance. It is not difficult to find that although YOLOv5n is lighter and faster, the detection accuracy is too low compared with SF-YOLOv5. Compared with YOLOv7-tiny, SF-YOLOv5 has advantages in detection accuracy, speed, and lightweight effect in the small object dataset, but it slightly increases the amount of calculation.

Because YOLOv3, YOLOv7, and ResNeXt-CSP are not lightweight algorithms, it is difficult to directly compare with SF-YLOv5, so we adjusted the scaling coefficient of SF-YOLOv5. This operation is adopted in YOLOv4, YOLOv5, and YOLOv7. The default value (0.33, 0.25) is adjusted to the same (1, 1) as YOLOv7. The adjusted SF-YOLOv5 was named SF-YOLOv5L. It has the same network structure and improvement idea as SF-YOLOv5, and the only difference is in the model size. The comparison of these algorithms is shown in Table 3. On the whole, SF-YOLOv5L is better than YOLOv3, ResNeXt-CSP, and its performance is close to that of the latest YOLOv7. This proved the feasibility and reliability of the improved method proposed in this paper.

### 4.7. Performance of Novel SF-YOLOv5 on Other Datasets

In this section, we further prove the ability of SF-YOLOv5 on other datasets, so as to judge whether the improvement made by SF-YOLOv5 in the direction of small object detection is universal.

TinyPerson [35], VisDrone [36], and VOC2012 [37,38] public datasets were selected for testing experiment in this section. Among them, TinyPerson and VisDrone contain a large number of small targets, which is suitable for the verification of SF-YOLOv5 proposed in this paper. Because there are not only small targets, but also a large number of large targets in VOC2012, it is mainly used to verify the performance of improved methods 2 and 3 proposed in this paper in the full-scale detection; therefore, in the experiment for VOC2012, we fine tuned the prediction head of SF-YOLOv5 and re added the P5 detection head. The algorithm is named SF-YOLOv5-P5. Table 4 shows the performance comparison of SF-YOLOv5 and YOLOv5s on these datasets.

To sum up, SF-YOLOv5 still achieved improvement in detection accuracy and comprehensive performance on TinyPerson, VisDrone and VOC2012 datasets. It is proved that the improved methods 1, 2, and 3 have obvious improvement effect in the direction for small object detection and have certain versatility in full-scale detection. Other experimental results in this section are shown in Figure 13 and Figure 14.

## 5. Conclusions

In this paper, a lightweight small object detection algorithm based on an improved feature fusion mode is proposed, which is dedicated to improving the detection effect on small targets. When the algorithm is significantly lighter, the detection accuracy is also significantly improved. By reducing the convolution down sampling operation in the network structure and trimming the output feature map, the model parameter amount and computation amount are significantly reduced under the premise of ensuring the detection accuracy. Ablation experiments show that the contribution of high-level feature maps generated by multiple convolution downsampling can be replaced by more effective feature fusion methods when detecting small objects. Besides this, in order to improve the small object detection accuracy of the model, a new feature fusion method (PB-FPN) is proposed based on PANet and BiFPN. By setting a fast SPP operation at the junction of the neck and the head, the feature expression potential of the output feature map of the backbone is fully exploited, and the detection performance of the algorithm is further improved. The revised model has good generalization ability and can also be applied to small object detection scenarios of other types of targets.

In the follow-up research, we will continue to improve the detection effect of the algorithm on ultra-small targets, and we will further explore the detection potential of the small object detection feature fusion method (PB-FPN) proposed in this paper on full-scale targets.

## Figures and Tables

**Figure 1 sensors-22-05817-f001:**
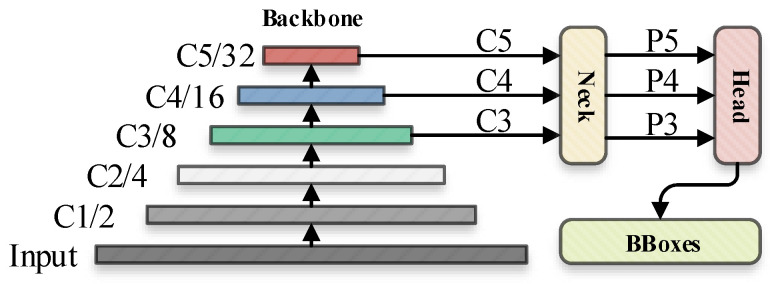
The default inference flowchart of YOLOv5.

**Figure 2 sensors-22-05817-f002:**
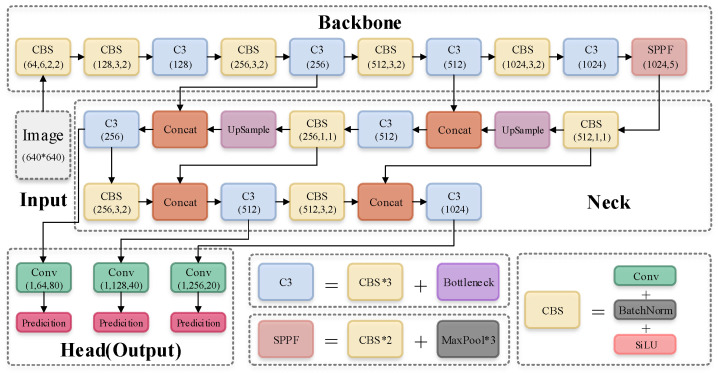
Default network structure of YOLOv5.

**Figure 3 sensors-22-05817-f003:**
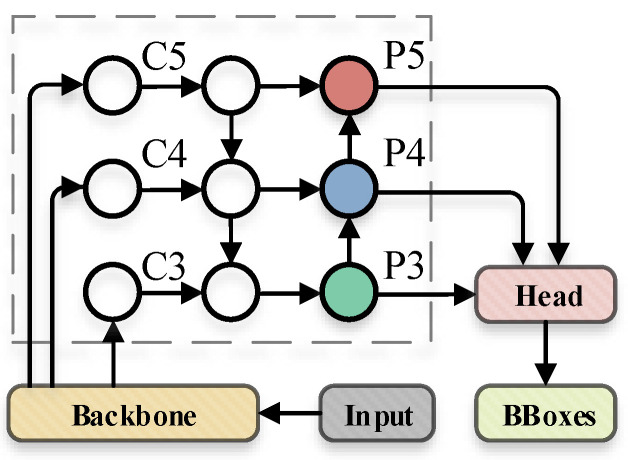
The dotted line in the figure is the default feature fusion path of YOLOv5.

**Figure 4 sensors-22-05817-f004:**
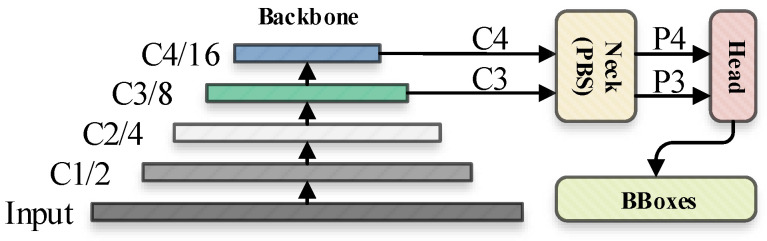
The inference flowchart of SF-YOLOv5. The backbone has been cut accordingly, corresponding to the improvement method proposed in Section 3.1, neck (PBS) represents a new feature fusion network with PB-FPN and SPPF prediction head, which corresponds to the methods described in Section 3.2 and Section 3.3 of the improved algorithm in this paper.

**Figure 5 sensors-22-05817-f005:**
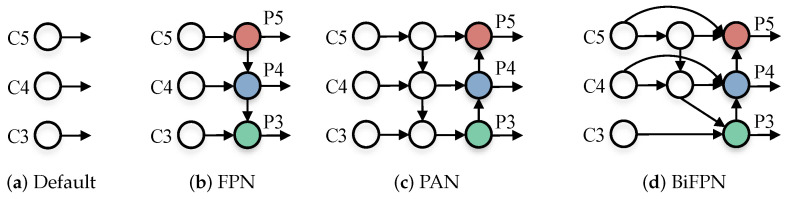
Some common feature fusion paths.

**Figure 6 sensors-22-05817-f006:**
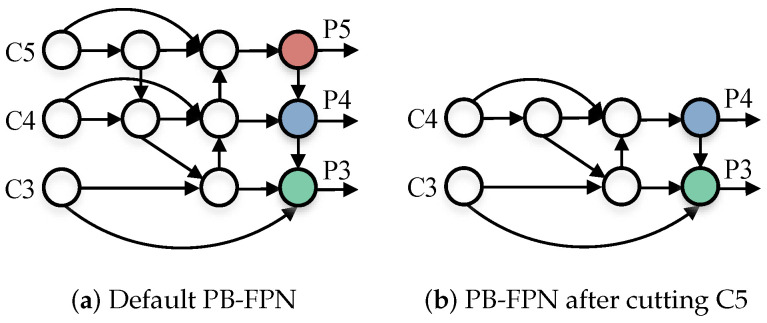
The feature fusion path of PB-FPN.

**Figure 7 sensors-22-05817-f007:**
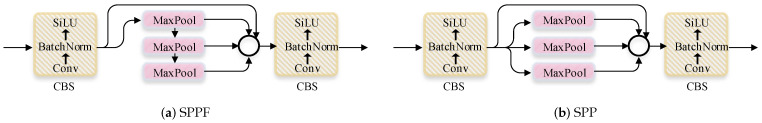
Structure comparison of SPPF and SPP.

**Figure 8 sensors-22-05817-f008:**
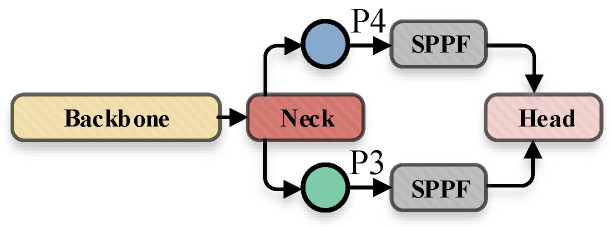
Structure diagram after adding SPPF at the junction of feature fusion network and prediction head.

**Figure 9 sensors-22-05817-f009:**
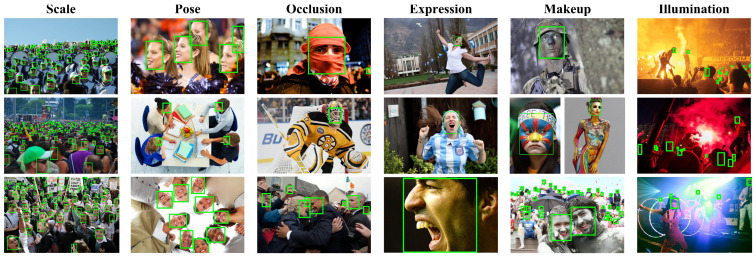
The WIDER FACE dataset contains a variety of complex and rich scenes [33].

**Figure 10 sensors-22-05817-f010:**
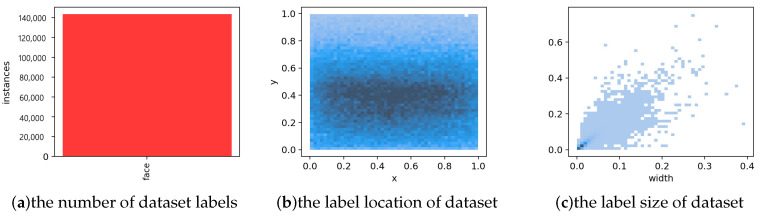
The attribute visualization results of the dataset used in this paper.

**Figure 11 sensors-22-05817-f011:**
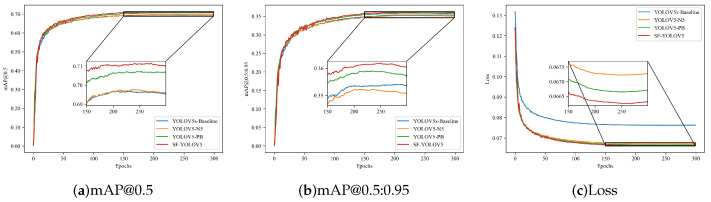
The ablation experimental results of the improved methods 1, 2, and 3.

**Figure 12 sensors-22-05817-f012:**
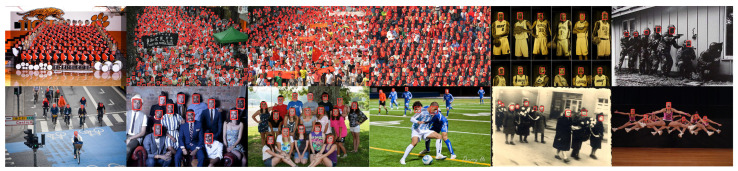
Detection effect of SF-YOLOv5 on WIDER FACE dataset.

**Figure 13 sensors-22-05817-f013:**
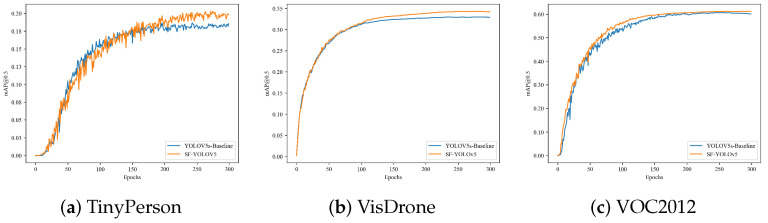
Comparison of detection accuracy between SF-YOLOv5 and YOLOv5s on other datasets.

**Figure 14 sensors-22-05817-f014:**
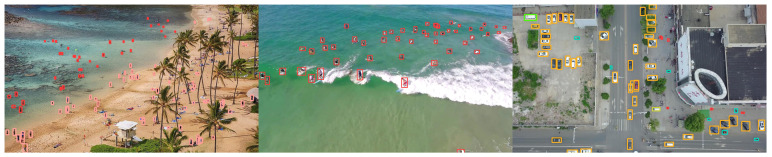
Detection effect of SF-YOLOv5 on other dataset.

**Table 1 sensors-22-05817-t001:** Comprehensive performance of five versions of YOLOv5 on COCO dataset [20].

Model	Size (Pixels)	mAP @0.5:0.95	mAP @0.5	Time CPU b1 (ms)	Time V100 b1 (ms)	Time V100 b32 (ms)	Params (M)	FLOPS @640 (B)
YOLOv5n	640	28.0	45.7	45	6.6	0.6	1.9	4.5
YOLOv5s	640	37.4	56.8	98	6.4	0.9	7.2	16.5
YOLOv5m	640	45.4	64.1	224	8.2	1.7	21.2	49.0
YOLOv5l	640	49.0	67.3	430	10.1	2.7	46.5	109.1
YOLOv5x	640	50.7	68.9	766	12.1	4.8	86.7	205.7

**Table 2 sensors-22-05817-t002:** Parameter of backbone in YOLOv5 network structure.

C	From	n	Params	Module	Arguments
0	−1	1	3520	CBS	[3, 32, 6, 2, 2]
1	−1	1	18,560	CBS	[32, 64, 3, 2]
2	−1	1	18,816	C3	[64, 64, 1]
3	−1	1	73,984	CBS	[64, 128, 3, 2]
4	−1	2	115,712	C3	[128, 128, 2]
5	−1	1	295,424	CBS	[128, 256, 3, 2]
6	−1	3	625,152	C3	[256, 256, 3]
7	−1	1	1,180,672	CBS	[256, 512, 3, 2]
8	−1	1	1,182,720	C3	[512, 512, 1]
9	−1	1	656,896	SPPF	[512, 512, 5]

**Table 3 sensors-22-05817-t003:** Performance comparison of SF-YOLOv5 and other algorithms on small object datasets.

Methods	Size	mAP@0.5	mAP@0.5:0.95	Parameters (M)	FLOPs (G)	Inference Time (ms)
YOLOv5s	640	69.7	35.5	7.01	15.8	13.1
YOLOv5-N5	640	69.9	35.3	1.88	11.2	10.4
YOLOv5-PB	640	70.8	36.0	2.03	12.7	11.3
SF-YOLOv5	640	71.3	36.3	2.23	13.8	12.2
improvement	-	+1.6	+0.8	−68.2%	−12.7%	−6.9%
YOLOv5n	640	63.9	30.8	1.76	4.20	8.60
SF-YOLOv5	640	71.3	36.3	2.23	13.8	12.2
improvement	-	+7.4	+5.5	+26.7%	+228.6%	+41.9%
YOLOv7-tiny	640	68.4	33.0	6.01	13.0	13.9
SF-YOLOv5	640	71.3	36.3	2.23	13.8	12.2
improvement	-	+2.9	+3.3	−62.9%	+6.2%	−12.2%
YOLOv3	640	74.5	39.5	61.5	154.7	44.7
SF-YOLOv5L	640	75.1	39.7	15.5	91.2	49.4
improvement	-	+0.6	+0.2	−74.8%	−41.0%	+10.5%
YOLOv7	640	76.1	39.5	36.5	103.2	18.0
SF-YOLOv5L	640	75.1	39.7	15.5	91.2	49.4
improvement	-	−1.0	+0.2	−57.5%	−11.6%	+174.4%
ResNeXt-CSP	640	73.7	37.6	31.8	58.9	32.6
SF-YOLOv5L	640	75.1	39.7	15.5	91.2	49.4
improvement	-	+1.4	+2.1	−51.3%	+54.8%	+51.5%

**Table 4 sensors-22-05817-t004:** Performance comparison between YOLOv5s and SF-YOLOv5 on other datasets.

Dataset	Methods	Size	mAP@0.5	mAP@0.5:0.95	Parameters (M)	FLOPs (G)	Inference Time (ms)
TinyPerson	YOLOv5s	640	18.7	6.0	7.02	15.8	12.8
	SF-YOLOv5	640	20.0	6.5	2.23	13.8	11.0
	improvement	-	+1.3	+0.5	−68.2%	−12.7%	−14.1%
VisDrone	YOLOv5s	640	33.0	17.9	7.04	15.9	12.6
	SF-YOLOv5	640	34.3	18.2	2.24	13.8	11.5
	improvement	-	+1.3	+0.3	−68.2%	−13.2%	−8.7%
VOC2012	YOLOv5s	640	60.8	37.0	7.06	15.9	25.8
	SF-YOLOv5-P5	640	61.2	38.3	4.59	15.7	24.8
	improvement	-	+0.4	+1.3	−34.9%	−1.3%	−3.9%

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
