# Peer review of "SF-YOLOv5: A Lightweight Small Object Detection Algorithm Based on Improved Feature Fusion Mode"

_sensors, 2022, doi:10.3390/s22155817_

Round 1
Reviewer 1 Report
The paper presents a modification for the one-stage detection algorithm YOLOv5 by clipping the feature map. That modification is motivated under the assumption of densely populated input images with small objects.
Although the work does not obtain notable to the original algorithm, the proposed method outlined a procedure to increase its performance under the working assumptions. The work is well presented and in a readable form.
The manuscript could be accepted for publication after minor revision of English. Moreover, I suggest modifying Tables 3 & 4. The modification should include dispersion measures (like standard deviation) for performance measures and eliminate the absolute improvement for each magnitude.
Author Response
To Whom It May Concern:
We are very thankful to the Editor-in-Chief, Associate Editor, and reviewers who have made critical comments and constructive suggestions concerning the work reported in our manuscript.
The manuscript has now been revised by taking into account these comments and suggestions. More details and experimental results are supplemented to test the new proposed algorithm and validated its effectiveness and performance. On behalf of my co-authors, I am now submitting the revised manuscript and our reply to the associated editor and reviewers as a supporting document.
Sincerely,
Haiying liu
Response to the Reviewer 1’s Comments:
- English language and style are fine/minor spell check required.
We read through the whole paper and carefully revised the English expression and language.
- I suggest modifying Tables 3 & 4. The modification should include dispersion measures (like standard deviation) for performance measures and eliminate the absolute improvement for each magnitude.
Thanks for the reviewer’s advice and we have modified the Tables 3 & 4. In the new version, we merged Tables 3 & 4 while eliminating the absolute improvement for each magnitude. At the same time, we revised the table style to make the results more intuitive.
For the dispersion measures (like standard deviation), because the experiment in this paper is based on computer simulation, the results are consistent under the same parameter settings. At the same time, we have checked a large number of papers (including our references) and found that their experiments do not use the dispersion measures. So we didn't make adjustments at this point. However, we will still seriously consider your follow-up suggestions.
Reviewer 2 Report
The paper provides an improved detection algorithm for small objects based on YOLOv5. Analyzing the paper, I identified the following issues:
1. English needs serious polishing. Some parts of the manuscript are hard to read and understand. There are numerous grammar mistakes and typos. Some examples (only from Abstract): a) line 2: it is written “object” instead of “objects”; b) line 7: it is written “model become” instead of “model becomes”; c) Two sentences are finished with semicolon (“;”) instead of full stop (“.”) – see lines 7 and 9; d) line 14: it is written “the parameters of the network was reduced” instead of “the parameters of the network were reduced”; e) line 15: it is written “computational resources (FLOPs) was reduced” instead of “computational resources (FLOPs) were reduced”. Please check the entire manuscript.
2. The experimental part (Section 4) is unconvincing. The method needs to be tested on other types of small objects (in the paper, the authors used only two databases with only faces). Some examples of databases that can be used are presented in Table 3 of the paper with doi: 10.1016/j.imavis.2020.103910. Moreover, the comparison needs to include other algorithms, outside the YOLO family.
3. Line 69: real-time cannot be improved. It is either real-time or not real-time; Maybe the authors speak about computational speed.
4. Section 2 needs to briefly survey some other relevant algorithms that can be used for detecting small objects. It is important to show why they chose YOLOv5 as a candidate algorithm to be improved.
5. Line 201: it is written “Parameters” instead of “number of parameters”. This mistake is found in many places in the manuscript.
6. Why in Table 3, the authors do not present some other versions of YOLOv5 (e.g. YOLOv5n) or even YOLOv6? Table 3 and 5 can be concatenated in a single Table.
7. “Author Contributions” section (see Journal’s Instructions for Authors) is not presented at the end of the paper.
Author Response
To Whom It May Concern:
We are very thankful to the Editor-in-Chief, Associate Editor, and reviewers who have made critical comments and constructive suggestions concerning the work reported in our manuscript.
The manuscript has now been revised by taking into account these comments and suggestions. More details and experimental results are supplemented to test the new proposed algorithm and validated its effectiveness and performance. On behalf of my co-authors, I am now submitting the revised manuscript and our reply to the associated editor and reviewers as a supporting document.
Sincerely,
Haiying liu
Response to the Reviewer 2’s Comments:
- English needs serious polishing. Some parts of the manuscript are hard to read and understand.
According to your suggestion, we have revised the paper and corrected all of the errors. At the same time, we checked the writing of the whole article.
- The experimental part (Section 4) is unconvincing. The method needs to be tested on other types of small objects (in the paper, the authors used only two databases with only faces). Some examples of databases that can be used are presented in Table 3 of the paper with doi: 10.1016/j.imavis.2020.103910. Moreover, the comparison needs to include other algorithms, outside the YOLO family.
We added two datasets (Visdrone and VOC2012) in the experiment section. Referring to the paper which you provided, we added the VOC2012 dataset. Thanks for your professional suggestions for the paper’s improvement. In the last part, we have plus this article as a reference.
At present, the experimental data of this paper has been supported by four public datasets. And we added a new comparison algorithm (ResNeXt) which is outside the YOLO family.
- Line 69: real-time cannot be improved. It is either real-time or not real-time; Maybe the authors speak about computational speed.
Thanks for the correction and we changed ‘real-time’ to ‘speed’ in the new version. Other parts of the paper have also been modified accordingly.
- Section 2 needs to briefly survey some other relevant algorithms that can be used for detecting small objects. It is important to show why they chose YOLOv5 as a candidate algorithm to be improved.
In the new version, we have investigated and summarized more current common classical algorithms, and offered the reasons for why we choosing YOLOv5 as a candidate algorithm.
- Line 201: it is written “Parameters” instead of “number of parameters”. This mistake is found in many places in the manuscript.
In the revised version, We checked the whole paper and revised ‘parameters’ to“number of parameters”. But the ‘Parameters(M)’ in the paper means the number of parameters in the model, so we have not modified the ‘Parameters(M)’.
- Why in Table 3, the authors do not present some other versions of YOLOv5 (e.g. YOLOv5n) or even YOLOv6? Table 3 and 5 can be concatenated in a single Table.
Thanks for your advices and we have merged Table 3 and 5 in the current version, and YOLOv5n was resented in the new table (Table 3). And in the new version, we supplemented the latest YOLOv7 and YOLOv7-tiny.
In the initial version, we didn't add YOLOv6 or YOLOv7 because they had not been released at that time. But at present, the latest YOLOv7 has been released. Compared with YOLOv6, it has higher performance and richer data support, so we chose YOLOv7 as the new comparison algorithm.
At the same time, during our experiment, we found that there are still some problems in the code of current YOLOv6, which will slow down the training speed. We are not sure whether this will affect the experimental results, which is also the reason why we didn't add YOLOv6 in the new version.
Yolov6 is also an excellent algorithm. At present, the official is also constantly updating and solving bugs, and we will continue to pay attention to its improvement in the future.
- Author Contributions” section (see Journal’s Instructions for Authors) is not presented at the end of the paper.
The Author Contributions” section now appears at the end of the paper.
Round 2
Reviewer 2 Report
The authors have successfully solved all my previous comments and concerns.